# Components of a Neanderthal gut microbiome recovered from fecal sediments from El Salt

Simone Rampelli [1,17], Silvia Turroni [1,17], Carolina Mallol[2,3,4], Cristo Hernandez[2], Bertila Galván[2], Ainara Sistiaga[5,6], Elena Biagi[1], Annalisa Astolfi[7,16], Patrizia Brigidi[8], Stefano Benazzi[9,10], Cecil M. Lewis Jr.[11,12], Christina Warinner [12,13], Courtney A. Hofman [11,12], Stephanie L. Schnorr [14,15,18 ✉] & Marco Candela [1,18 ✉]

A comprehensive view of our evolutionary history cannot ignore the ancestral features of our gut microbiota. To provide some glimpse into the past, we searched for human gut microbiome components in ancient DNA from 14 archeological sediments spanning four stratigraphic units of El Salt Middle Paleolithic site (Spain), including layers of unit X, which has yielded well-preserved Neanderthal occupation deposits dating around 50 kya. According to our findings, bacterial genera belonging to families known to be part of the modern human gut microbiome are abundantly represented only across unit X samples, showing that well-known beneficial gut commensals, such as *Blautia*, *Dorea*, *Roseburia*, *Ruminococcus*, *Faecalibacterium* and *Bifidobacterium* already populated the intestinal microbiome of *Homo* since as far back as the last common ancestor between humans and Neanderthals.

[1] Unit of Microbiome Science and Biotechnology, Department of Pharmacy and Biotechnology, University of Bologna, Via Belmeloro 6, Bologna, Italy. [2] Department of Geography and History, University of La Laguna, Campus de Guajara, La Laguna, Tenerife, Spain. [3] Archaeological Micromorphology and Biomarker Research Lab, University of La Laguna, Avenida Astrofísico Francisco Sánchez 2, La Laguna, Tenerife, Spain. [4] ICArEHB - Interdisciplinary Center for Archaeology and the Evolution of Human Behaviour, Universidade do Algarve, Campus de Gambelas, Edificio 1, Faro, Portugal. [5] Earth, Atmospheric and Planetary Sciences Department, Massachusetts Institute of Technology, 77 Massachusetts Ave, Cambridge, MA, USA. [6] GLOBE Institute, Faculty of Health and Medical Sciences, University of Copenhagen, Oester Voldgade 5-7, Copenhagen, Denmark. [7] "Giorgio Prodi" Cancer Research Center, University of Bologna, Via Massarenti 11, Bologna, Italy. [8] Department of Medical and Surgical Sciences, University of Bologna, Via Massarenti 9, Bologna, Italy. [9] Department of Cultural Heritage, University of Bologna, Via degli Ariani 1, Ravenna, Italy. [10] Department of Human Evolution, Max Planck Institute for Evolutionary Anthropology, Deutscher Platz 6, Leipzig, Germany. [11] Laboratories of Molecular Anthropology and Microbiome Research, University of Oklahoma, 101 David L. Boren Blvd, Norman, OK, USA. [12] Department of Anthropology, University of Oklahoma, 455W Lindsey St, Norman, OK, USA. [13] Department of Archaeogenetics, Max Planck Institute for the Science of Human History, Kahlaische Strasse 10, Jena, Germany. [14] Konrad Lorenz Institute for Evolution and Cognition Research, Martinstraße 12, Klosterneuburg, Austria. [15] Department of Anthropology, University of Nevada, 4505S. Maryland Pkwy, Las Vegas, NV, USA. [16]Present address: Department of Morphology, Surgery and Experimental Medicine, University of Ferrara, Via Fossato di Mortara 70, Ferrara, Italy. [17]These authors contributed equally: Simone Rampelli and Silvia Turroni. [18]These authors jointly supervised this work: Stephanie L. Schnorr and Marco Candela. ✉email: stephanie.schnorr@kli.ac.at; marco.candela@unibo.it

Over the past decade, microbiome research has high-lighted the crucial role that the gut microbiome plays in human biology through its pleiotropic influence on many physiological functions, such as human development, immunity, metabolism and neurogenerative processes[1]. This body of knowledge has catalyzed interest in incorporating the gut microbiome into our evolutionary history, as an adaptive partner providing the necessary phenotypic plasticity to buffer dietary and environmental changes. Studies aimed at exploring the ancestral traits of the human gut microbiome are therefore encouraged, as a unique evolutionary perspective to improve our knowledge of gut microbiome assembly and interactions with the human host[2].

The ancestral configuration of the human gut microbiome has generally been inferred by microbiome data stemming from contemporary populations found across all six human occupied continents who adhere to traditional lifestyles, such as the Hadza hunter-gatherers from Tanzania, the rural Bassa from Nigeria and rural Papuans from Papua New Guinea, among others[3–11]. However, since this research involved modern populations, no direct information on the ancient human gut microbiome structure can actually be provided.

Alternatively, ancient DNA (aDNA) analysis based on shot-gun metagenomic sequencing is emerging as an attractive and reliable opportunity to directly investigate the microbial ecology of our ancestors[12–15]. Paleomicrobiological aDNA studies have traditionally been conducted on dental calculus and bones[15–18], providing landmark information on ancient pathogens and oral microbial communities. However, given that stools are widely acknowledged as a proxy of the gut microbiome structure[19], the metagenomics of aDNA from paleofeces, also known as coprolites, represents the only way to gain insight into the ancient human gut microbiome[2]. Pioneering studies in this direction have been carried out, providing next-generation sequencing data from modern human mummified intestinal contents and paleofeces[20–24]. Nevertheless, to the best of our knowledge, paleofecal samples older than 8,000 years have never being analyzed, leaving an important gap on the pre-historical human gut microbiome configuration.

In this scenario, we attempted to identify ancient human gut microbiome components by shotgun metagenomic analysis of aDNA extracted from archeological sedimentary samples (ES1 to ES7) from the stratigraphic unit (SU) X (subunit Xb-H44) of the Middle Paleolithic open-air site, El Salt (Alicante, Spain)[25] (Fig. 1). The archeological setting of El Salt yielded evidence of recurrent occupation by Neanderthals, our closest evolutionary relatives, dated between $60.7 \pm 8.9$ and $45.2 \pm 3.4$ kya[26,27]. In particular, the sedimentary samples ES1-7 have been previously shown to include several millimetric phosphatic coprolites and fecal lipid biomarkers, namely coprostanol and 5ß-stigmasta-nol, with proportions suggesting a human origin[25]. These samples therefore represent, to our knowledge, the oldest known positive identification of human fecal matter. The present work also includes an additional seven new archeological sediments collected in 2018 as a control. Two were from SU X (subunit Xa and Xb, respectively) and the others from sur-rounding SUs, i.e., upper V (three samples), IX and XI (one sample each) (Fig. 2a). While SUs IX to XI are associated with rich archaeological assemblages, upper SU V yielded very few archaeological remains[26]. We found that samples positive for the presence of fecal biomarkers showed traces of both ancient human mtDNA and ancient components of the modern human gut microbiome. These components included so-called "old friends" and beneficial commensal inhabitants of modern human guts, providing unique insights into their relevance to the biology of the *Homo* lineage.

## Results and discussion

**Ancient DNA sequencing and damage assessment.** DNA was extracted from 14 archeological sedimentary samples and pre-pared for shotgun metagenomics in a dedicated aDNA facility at the Laboratories of Molecular Anthropology and Microbiome Research in Norman (OK, USA) (see Methods). A total of 124,592,506 high-quality paired-end sequences were obtained by Illumina NextSeq sequencing and analyzed for bacterial aDNA. To remove contamination by modern DNA, which is one of the major complications in studies of ancient samples[14,28,29], we evaluated the DNA damage pattern as compared with present-day DNA references. In particular, Skoglund and colleagues[12] translated the pattern of cytosine deamination into a postmortem degradation score (PMDS), which provides information on whether a given sequence is likely to derive from a degraded aDNA molecule. Reads were aligned against all bacterial genomes of the NCBI database, and ancient bacterial reads were recovered by setting PMDS > 5, to minimize the probability of a sequence being from a present-day contaminating source[12]. An average of 6,836 sequences per sample (range, 279–17,901) were retained, corresponding to a small but consistent fraction of DNA being ancient and derived from bacteria (mean ± SD, 0.069% ± 0.029%) (Supplementary Table 1 and Supplementary Fig. 1). The same procedure was applied to extraction, library and PCR blanks, resulting in the retrieval of a minimal number of 144, 1, and 42 ancient bacterial sequences, respectively. Ancient reads from blanks were assigned to 116 bacterial species that showed no overlap with the sample dataset (Supplementary Data 1). When comparing the fraction of reads with PMDS > 5 per million reads between samples IX, Xa, ES1-7, Xb and XI (i.e., those positive for the presence of fecal biomarkers and/or associated with rich archaeological assemblages), and samples from SU V (i.e., with no or very few archeological remains), the first showed a greater abundance of PMDS > 5 reads (*p*-value = 0.01, Wilcoxon test) (Supplementary Fig. 2), possibly as a result of the presence of human fecal sediment.

**Detection of ancient human mitochondrial DNA.** In order to detect human aDNA traces in our sample set, we searched for human mitochondrial DNA (mtDNA) sequences in PMDS-filtered metagenomes obtained from the 14 archeological sedimentary samples. Ancient human mtDNA was detected in almost all ES1 to ES7 samples from SU X (Fig. 2b). No traces of mtDNA from other animals were detected. To strengthen these findings, all samples were subjected to target capture of mtDNA with a Neanderthal bait panel (Arbor Biosciences; see Methods), and subsequent sequencing on Illumina NextSeq platform. Based on this analysis, ES1, ES2, ES5 and Xb samples tested positive for the presence of ancient human mtDNA, showing >1000 human mtDNA reads with PMDS > 1, breath of coverage >10%, −Δ % ≥ 0.9 and modern contamination less than 2% (Fig. 2b and Supplementary Table 2)[30]. Taken together, this evidence strongly supports human origin for the El Salt samples, particularly those from SU X[25].

**Profiling of ancient prokaryotic DNA.** As for prokaryotic aDNA, seventeen bacterial and one archaeal phyla were identified in the aDNA sedimentary record of El Salt, with different representation across SUs (Fig. 3). As expected for its wide dis-tribution in nature[31,32], Actinobacteria is the most represented phylum, with environmental species from *Streptomycetaceae*, *Pseudonocardiaceae*, *Micromonosporaceae*, *Nocardiaceae*, *Myco-bacteriaceae*, *Microbacteriaceae* and *Nocardioidaceae* families being detected in almost all the SUs. Similarly, the vast majority of sediment samples share a number of ancient sequences assigned to *Bacillaceae* members, which are known to play

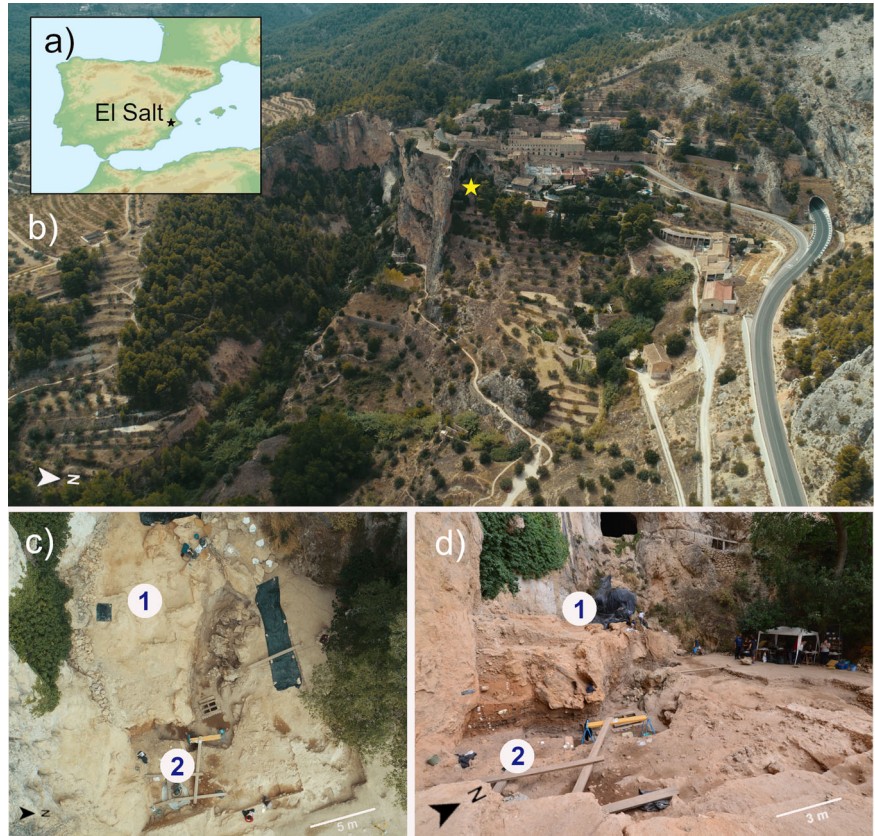

**Fig. 1 The Middle Paleolithic site of El Salt (Spain). a, b** General site setting. The yellow star marks the location of the excavated area. As can be seen in the photograph, the sedimentary deposit rests against a tall limestone wall. **c, d** Different views of the excavation area indicating the zones sampled for this study. Zone 1 includes samples V1-3 and Zone 2 all the rest.

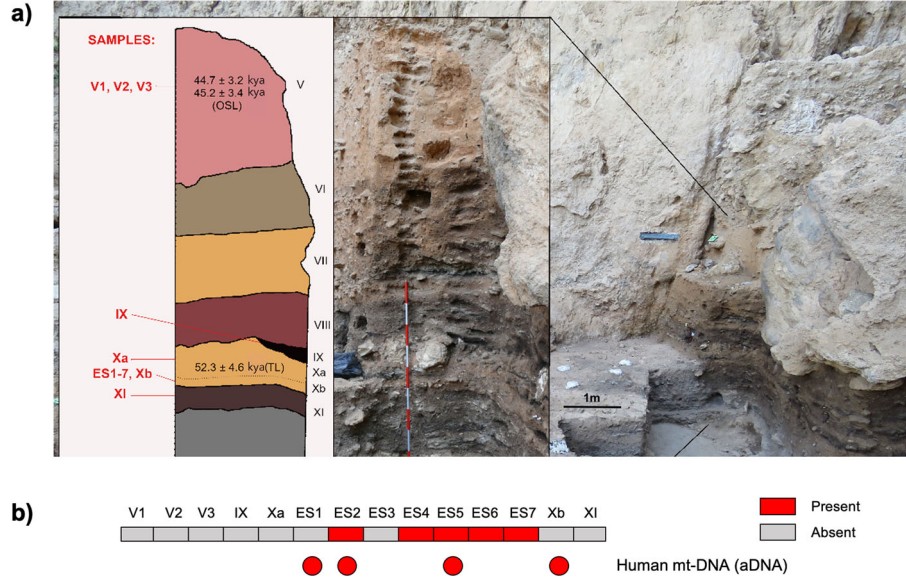

**Fig. 2 Pleistocene stratigraphic sequence of El Salt (Units V-XI), chronometric dates of the sampled units, and evidence of traces of ancient human mitochondrial DNA. a** The 14 sediment samples included in this study are shown in red. Samples ES1 to ES7 (subunit Xb-H44) are from Sistiaga et al.[25]. **b** Red boxes, human mtDNA fragments as recovered from metagenomic sequencing data; red circles, confirmation of the presence of ancient human mtDNA through target capture and sequencing (please, see Methods for further details).

fundamental roles in soil ecology, where they can persist up to thousands of years, if not longer, due to their ability to form resistant endospores[33,34]. Another large fraction of aDNA shared by most SUs includes Proteobacteria constituents, especially from Alphaproteobacteria (mainly *Rhodobacteraceae*, *Rhodospirillaceae* and *Sphingomonadaceae* families), Betaproteobacteria (mainly *Comamonadaceae* and *Burkholderiaceae*) and Gammaproteobacteria (with *Xanthomonadaceae*) classes. Again, these are

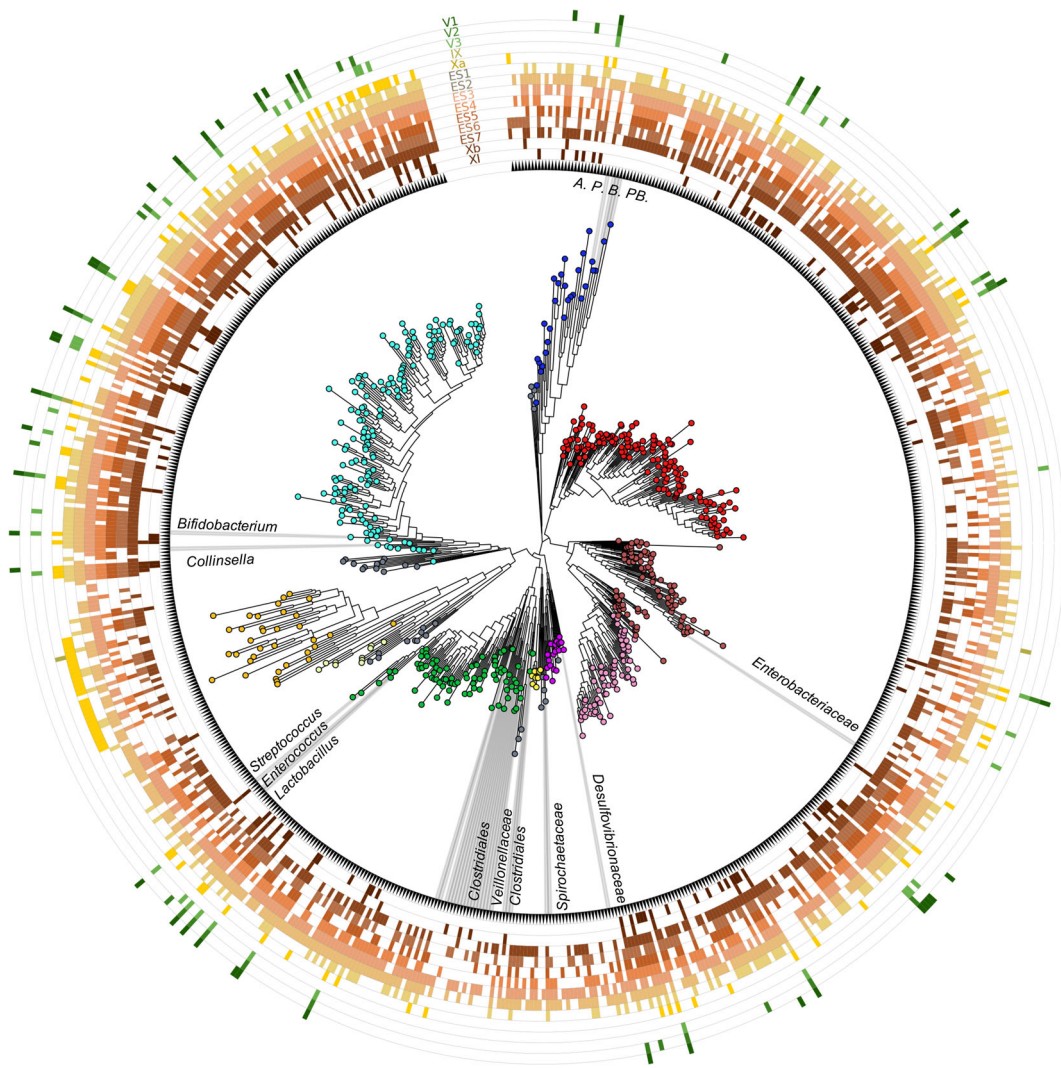

**Fig. 3 Ancient bacteria in sediment samples from El Salt.** The phylogenetic tree was built with representative sequences of bacterial genera for which at least one species was present with more than four hits in one sample. Different colors indicate different phyla (or classes for Proteobacteria) as follows: cyan, Actinobacteria; blue, Bacteroidetes; red, Alphaproteobacteria; brown, Gammaproteobacteria; pink, Betaproteobacteria; purple, Deltaproteobacteria; yellow, Acidobacteria; green, Firmicutes; light-yellow, Planctomycetes; orange, Euryarchaeota; grey, others (including Armatimonadetes, Chlorobi, Chloroflexi, Cyanobacteria, Deinococcus-Thermus, Fusobacteria, Gemmatimonadetes, Nitrospirae, Spirochaetes, Synergistetes and Verrucomicrobia). Bacterial taxa belonging to families common to the gut microbiome of hominids are highlighted at different taxonomic level. A, *Alistipes*; P, *Prevotella*; B, *Bacteroides*; PB, *Parabacteroides*.

cosmopolitan bacteria commonly found in both terrestrial and aquatic environments, as free-living organisms or symbionts in different hosts[35,36]. In light of their DNA damage pattern, it is reasonable to assume that these are truly ancient environmental bacteria that populated archeological sediments. The contamination of archeological remains by environmental bacteria is indeed well expected, as already documented in previous paleomicrobiological aDNA studies[15,18,37]. For the relative abundances of bacterial families detected across the samples, please see Supplementary Data 2.

**Putative components of the Neanderthal gut microbiome.** Next, following an approach similar to Weyrich et al.[18], who first characterized the oral microbiome from Neanderthal dental calculus, we focused our analysis on intestinal microorganisms. Specifically, in order to identify potential ancient human gut microbiome components, we searched for bacterial genera belonging to the 24 families that have recently been indicated as being common to the gut microbiome of

hominids (i.e., *Methanobacteriaceae*, *Bifidobacteriaceae*, *Coriobacteriaceae*, *Bacteroidaceae*, *Porphyromonadaceae*, *Prevotellaceae*, *Rikenellaceae*, *Tannerellaceae*, *Enterococcaceae*, *Lactobacillaceae*, *Streptococcaceae*, *Christensenellaceae*, *Clostridiaceae*, *Eubacteriaceae*, *Lachnospiraceae*, *Oscillospiraceae*, *Peptostreptococcaceae*, *Ruminococcaceae*, *Erysipelotrichaceae*, *Veillonellaceae*, *Desulfovibrionaceae*, *Succinivibrionaceae*, *Enterobacteriaceae* and *Spirochaetaceae*)[38–45]. Accordingly, while harboring similar family-level gut microbiome profiles, humans and non-human hominids, including our closest living relatives—chimpanzees, can be differentiated on the basis of the particular pattern of associated gut microbiome genera (as well as species and strains) represented within these families[45]. This strong association between gut microbiome composition and host physiology—known as phylosymbiosis—is believed to be universal in mammals, essentially as a result of all the physical, chemical and immunological factors that differentiate the intestine of the host species (e.g., type of digestive organs, pH, oxygen level,

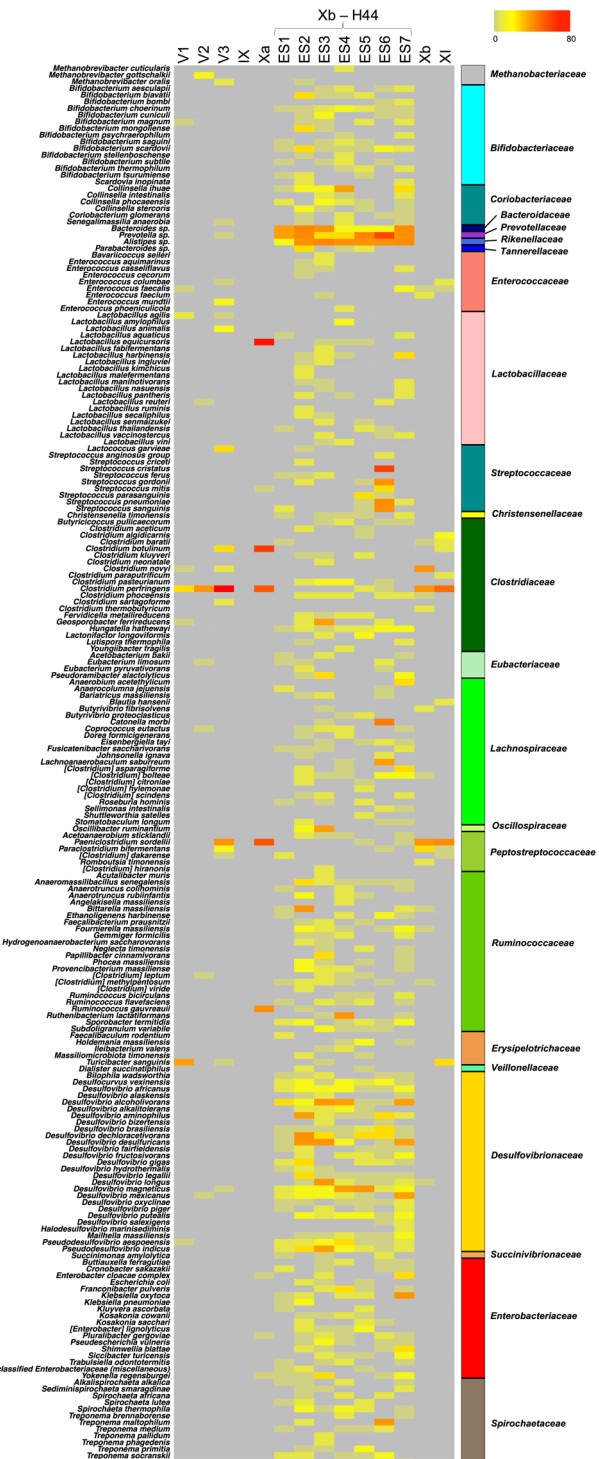

**Fig. 4 Ancient components of the human gut microbiome in sediment samples from El Salt.** The heat map shows the hit number distribution across sediment samples from the different stratigraphic units. Only ancient bacterial species with ≥ 2 hits in at least one sample were kept. See also Supplementary Data 3.

Fig. 3, we provide the overall compositional profile of the El Salt samples from SU IX, X and XI restricted to the hominid-associated gut microbiome families, while in Supplementary Fig. 4, the proportions of these families are compared with those of the samples with no or very few archeological remains (i.e., from SU V). The compositional profile of samples from SU IX, X and XI was next compared to publicly available gut microbiomes from contemporary human populations as representative of different subsistence practices, such as Hadza and Matses hunter-gatherers, Tunapuco rural agriculturalists and western urbans from Italy and the US[7,47]. As shown by the Principal Coordinates Analysis of Bray-Curtis distances between the family-level profiles (Supplementary Fig. 5), the El Salt samples from SU IX, X and XI tend to cluster closer to Tunapuco and Matses, resembling more the "ancestral" human gut microbiome of rural agriculturalists and hunter-gatherers than the urban western gut microbiome[7]. However, as the degree of degradation of microbial DNA in ancient samples might be different for various gut microbiome components, any conclusions from these compositional data must be taken with due caution.

Further supporting a human-host origin of the bacterial species belonging to the hominid-associated gut microbiome families detected in the El Salt samples from SU IX, X and XI, feces or gastrointestinal tract are the first documented isolation source for 91 species out of 210 (43.3%), with 60 of these being classifiable as closely related to the human gut (Supplementary Data 3). In the latter subgroup, we can count several species from Lachnospiraceae (including well-known (beneficial) commensal inhabitants of modern human guts, such as *Blautia*, *Coprococcus*, *Dorea*, *Fusicatenibacter* and *Roseburia* spp.) and *Ruminococcaeae* families. Particularly, within *Ruminococcaceae*, we detected members of *Anaerotruncus*, *Ruminococcus* and *Subdoligranulum* genera, and the butyrate producer *Faecalibacterium*, one of the human commensal bacteria of greatest current interest, due to its very promising potential as a biomarker of a healthy gut microbiome[48]. Most of the aforementioned bacterial genera have been reported to account for the phylotypic diversity between human and non-human hominids, showing strong bias towards a human-host[45]. It is also worth remembering that most of these bacteria are able to produce short-chain fatty acids (mainly acetate and butyrate) from the fermentation of indigestible carbohydrates, through the establishment of complex syntrophic networks. Short-chain fatty acids are today considered metabolic and immunological gut microbiome players with a leading role in human physiology[49]. In addition, the Xb-H44 samples showed a high number of hits for *Bacteroides*, *Parabacteroides*, *Alistipes* and *Bifidobacterium* spp., other genera known to prevail in the human gut microbiome[39,45]. Interestingly, *Bacteroides* and *Bifidobacterium* have been shown to exhibit patterns of co-speciation with hominids[45]. For *Bifidobacterium*, this is particularly consistent with the propensity of this genus to be maternally inherited across generations. Being capable of metabolizing milk oligosaccharides and acting as a potent immunomodulator, the presence of vertically transmitted *Bifidobacterium* spp. in the infant gut could have provided important growth benefits to infant Hominidae[50–52].

To further characterize the ancient microbial taxa detected in the El Salt samples, we applied the HOPS[53]-based approach recently used by Jensen et al.[30]. In short, all the reads were first annotated and, subsequently, the ancient origin of each taxon was authenticated by computing three indicators: (i) the fraction of reads with PMDS > 1, (ii) the negative difference proportion (−Δ %) of PMDS > 1 reads, and (iii) their deamination rate at 5′. Taxa showing more than 200 assigned reads, more than 50 reads with PMDS > 1, −Δ % = 1 and C-T transition at 5′ >10% were

host-derived molecules and immune system)[46]. According to our findings, 210 bacterial species belonging to hominid-associated gut microbiome families, as listed above, are represented in the aDNA from El Salt SU IX, X and XI, with the highest detection rate in samples from SU X and, particularly, in ES1 to ES7 (Fig. 4), for which a human-like host origin had been previously suggested[25]. In Supplementary

**Table 1 List of the 36 most abundant microbial taxa identified in the El Salt sediments, belonging to the hominid gut microbiome families.**

| Species | Reads | Reads with PMDS > 1 | DoC | >1x (%) | C-T 5′ (%) | −Δ % |
|---|---|---|---|---|---|---|
| *Alistipes finegoldi* | 340 | 80 | 0.003 | 0.3 | 11.3 | 1 |
| *Alistipes shahii* | 338 | 68 | 0.003 | 0.3 | 11.9 | 1 |
| *Alistipes indistinctus* | 245 | 62 | 0.003 | 0.2 | 11.4 | 1 |
| *Alistipes timonensis* | 303 | 55 | 0.003 | 0.3 | 11.1 | 1 |
| *Alistipes senegalensis* | 376 | 73 | 0.003 | 0.3 | 13.4 | 1 |
| *Alistipes ihumii* | 406 | 93 | 0.005 | 0.4 | 10.9 | 1 |
| *Anaeromassilibacillus senegalensis* | 233 | 54 | 0.002 | 0.2 | 10.2 | 1 |
| *Bifidobacterium callitrichos* | 322 | 65 | 0.003 | 0.3 | 10.9 | 1 |
| *Bifidobacterium subtile* | 288 | 54 | 0.003 | 0.3 | 11.7 | 1 |
| *Bittarella massiliensis* | 474 | 110 | 0.006 | 0.5 | 10.5 | 1 |
| *Catonella morbi* | 543 | 80 | 0.008 | 0.8 | 10.5 | 1 |
| *Clostridium perfringes* | 1392 | 194 | 0.03 | 0.7 | 12.6 | 1 |
| *Collinsella ihuae* | 513 | 106 | 0.006 | 0.5 | 10.2 | 1 |
| *Collinsella phocaeensis* | 386 | 83 | 0.003 | 0.3 | 11.6 | 1 |
| *Desulfovibrio alaskensis* | 286 | 51 | 0.002 | 0.2 | 10.2 | 1 |
| *Desulfovibrio alkalitolerans* | 652 | 124 | 0.007 | 0.6 | 10.3 | 1 |
| *Desulfovibrio dechloracetivorans* | 608 | 122 | 0.005 | 0.4 | 10.6 | 1 |
| *Desulfovibrio fairfieldensis* | 383 | 74 | 0.003 | 0.3 | 12.1 | 1 |
| *Desulfovibrio gracilis* | 289 | 56 | 0.003 | 0.3 | 11.1 | 1 |
| *Desulfovibrio legallii* | 374 | 78 | 0.004 | 0.4 | 10.2 | 1 |
| *Desulfovibrio oxyclinae* | 356 | 84 | 0.003 | 0.3 | 10.3 | 1 |
| *Desulfovibrio vulgaris* | 668 | 133 | 0.005 | 0.5 | 10.5 | 1 |
| *Faecalibacterium prausnitzii* | 287 | 61 | 0.003 | 0.3 | 12.7 | 1 |
| *Fournierella massiliensis* | 313 | 68 | 0.003 | 0.2 | 11.1 | 1 |
| *Hungatella (Clostridium) hatheway* | 276 | 53 | 0.001 | 0.1 | 10.4 | 1 |
| *Klebsiella pneumoniae* | 390 | 82 | 0.002 | 0.2 | 10.9 | 1 |
| *Paeniclostridium sordelli* | 780 | 86 | 0.01 | 0.5 | 10.5 | 1 |
| *Papillibacter cinnamivorans* | 284 | 53 | 0.003 | 0.3 | 11.3 | 1 |
| *Prevotella timonensis* | 220 | 96 | 0.004 | 0.1 | 15.9 | 1 |
| *Prevotella saccharolytica* | 494 | 61 | 0.008 | 0.8 | 12.1 | 1 |
| *Pseudoescherichia vulneris* | 278 | 56 | 0.002 | 0.2 | 12.2 | 1 |
| *Pseudosulfovibrio indicus* | 605 | 123 | 0.005 | 0.5 | 10.3 | 1 |
| *Rikenella microfusus* | 276 | 54 | 0.003 | 0.3 | 12.0 | 1 |
| *Ruthenibacterium lactatiformans* | 315 | 57 | 0.002 | 0.2 | 10.1 | 1 |
| *Siccibacter turicensis* | 311 | 58 | 0.002 | 0.2 | 11.2 | 1 |
| *Yokenella regensburgei* | 258 | 56 | 0.002 | 0.1 | 11.8 | 1 |

Depth (DoC) and breadth of coverage (>1x) were calculated using BEDTools. Deamination rates at the 5′ ends of DNA fragments were calculated for the first 10 bases using mapDamage. −Δ % refers to the negative difference proportion introduced by Hübler et al.[53]. C-T (%) and −Δ % are computed on PMDS>1 reads.

considered to be of ancient origin (see Table 1 and Supplementary Figs. 6–8 for MapDamage plots, coverage plots and edit distance distribution)[30]. This in-depth characterization of the microbial metagenomic reads from the El Salt samples allowed us to confirm the presence of several species belonging to the gut microbiome families of hominids (including, among others, *Alistipes*, *Bifidobacterium*, *Desulfovibrio* and *Prevotella* spp., and *Faecalibacterium prausnitzii*), showing a read profile consistent with their ancient origin.

As mentioned above, high amounts of coprostanol, a metabolite formed through hydrogenation of cholesterol by specific bacteria in the intestine of higher mammals, were found in some of the El Salt sediments from SU X, with proportions consistent with the presence of human fecal matter[25]. We therefore specifically looked for microorganisms capable of this metabolism in the aDNA from El Salt samples. To date, cholesterol-reducing capabilities associated with coprostanol conversion in feces have been suggested for *Bifidobacterium*, *Collinsella*, *Bacteroides*, *Prevotella*, *Alistipes*, *Parabacteroides*, *Enterococcus*, *Lactobacillus*, *Streptococcus*, *Eubacterium*, *Lachnospiraceae* (e.g., *Coprococcus* and *Roseburia*) and *Ruminococcaceae* (e.g., *Anaerotruncus*, *Faecalibacterium*, *Ruminococcus* and *Subdoligranulum*)[54–58], which were all detected, at variable but

substantial abundances, within the species belonging to the 24 gut microbiome families (as defined above) in Xa and Xb-H44 subunits. While lending support to the presence of coprostanol in the same layer as reported by Sistiaga et al.[25], our findings on the representation of potential cholesterol-reducing bacteria in Neanderthal feces point to the microbial metabolism of cholesterol as an important function of the human gut microbiome for both modern and ancient humans, and suggest that relatively higher cholesterol intake has been a feature of the human diet at least since the Middle Pleistocene.

Finally, the remaining bacterial species belonging to the hominid gut microbiome families identified in El Salt sediments from SUs IX, X and XI could be sorted into two major source categories: human (or animal) oral and/or pathobiont, and environmental (see Supplementary Fig. 9). In particular, possibly consistent with evidence of dental caries and periodontal disease in Neanderthals[18], we found traces of potential opportunistic pathogens (e.g., *Methanobrevibacter oralis*, *Scardovia inopinata*, *Streptococcus parasanguinis*, *Streptococcus sanguinis*, *Pseudoramibacter alactolyticus*, *Catonella morbi*, *Johnsonella ignava*, *Lachnoanaerobaculum saburreum*, *Shuttleworthia satelles*, *Stomatobaculum longum*, *Treponema maltophilum*, *Treponema medium*, *Treponema socranskii* and

*Treponema vincentii*), which have been associated with modern oral and dental diseases in humans[59–68].

Expectedly, the samples from the upper part of SU V (which are poor in archaeological remains) showed scarce and inconsistent presence of aDNA related to hominid-associated gut microbiome bacterial families. The highest hit counts were found for *Clostridium perfringens*, *Paeniclostridium sordellii* and *Turicibacter sanguinis*, with the first two being environmental opportunistic microorganisms historically associated with human gangrene and the last with acute appendicitis[69–71]. These findings further support the presence of potential human-like gut microbiome components as being unique to the samples from Xa and Xb, the only sedimentary layers that to date have shown traces of microscopic coprolites and fecal lipid biomarkers of presumed archaic human origin.

In conclusion, by reconstructing ancient bacterial profiles from El Salt Neanderthal feces-containing sediments, we propose the existence of a core human gut microbiome with recognizable coherence between Neanderthals and modern humans, whose existence would pre-date the split between these two lineages, i.e., in the early Middle Pleistocene[72]. Although the risk of fractional contamination by modern DNA can never be ruled out and our data must be taken with some caution, the identification of this ancient human gut microbiome core supports the existence of evolutionary symbioses with strong potential to have a major impact on our health. In particular, the presence of known short-chain fatty acid producers, such as *Blautia*, *Dorea*, *Roseburia*, *Rumunicoccus*, *Subdoligranulum*, *Faecalibacterium* and *Bifidobacterium*, among the gut microbiome of Neanderthals, provides a unique perspective on their relevance as keystone taxa to the biology and health of the *Homo* lineage. While the former are known to allow extra energy to be extracted from dietary fiber[73], strengthening the relevance of plant foods in human evolution, *Bifidobacterium* could have provided benefits to archaic human mothers and infants as a protective and immunomodulatory microorganism. Furthermore, the detection of so-called "old friend" microorganisms[74] as putative components of Neanderthal gut microbiome (e.g., *Spirochaetaceae*, *Prevotella* and *Desulfovibrio*) further supports the hypothesized ancestral nature of these human gut microbiome members, which are now disappearing in westernized populations[3–11]. In the current scenario where we are witnessing a wholesale loss of bacterial diversity in the gut microbiome of the cultural "west", with the parallel rise in dysbiosis-related autoimmune and inflammatory disorders[75], the identification of evolutionarily integral taxa of the human holobiont may benefit practical applications favoring their retention among populations living in or transitioning to increasingly microbially deplete contexts. Such therapeutic applications may in the near future include next-generation probiotics, prebiotics or other gut microbiome-tailored dietary interventions.

## Methods

**Site and sampling**. All samples used for this study were collected from the archaeological site of El Salt, Alicante, Spain. The archaeological team led by B. Galván conducted the excavations under a government permit and following the Spanish heritage law (No. 16/1985, 25 June). All excavated material including the sedimentary material is interpreted as archaeological material so no further permits are required for the presented study. Loose sediment samples (5–10 g) were collected in plastic vials using sterilized spoons (one per sample) after thoroughly cleaning the excavation surface with a vacuum cleaner in order to guarantee removal of any recent dust or sediment blown in from a different location. Lab safety masks and nitrile gloves were used at all times. The samples were collected from two different zones of the current El Salt excavation area (see Fig. 1):

1. Zone 1. This is the upper excavation zone. Samples were collected from SU V, Facies 23 (one sample, V1) and Facies 24 (two samples, V2 and V3). This unit has been dated by OSL to $44.7 \pm 3.2$ ky BP[76]. Lithologically, it is composed of massive, loose yellowish-brown calcareous silt with coarse sand and isolated larger limestone and travertine clasts. Facies 23 is fine-grained,

while Facies 24 (overlying Facies 23) also contains gravel. Unit V has yielded very few archaeological remains (bone fragments and technologically undiagnostic flint flakes).

2. Zone 2. This is the lower excavation zone. Samples were collected from SUs IX, Xa, Xb and XI, which are a stratified succession of sedimentary layers rich in Middle Paleolithic archaeological remains (charcoal, combustion features and burnt and unburnt bone and flint artifacts). From top to base:

Unit IX (one sample): is the uppermost layer in this succession. It is discontinuous across the excavation area, comprising a series of dark brown-black sandy silt lenses.
Unit Xa (one sample): dated by TL to $52.3 \pm 4.6$ ky BP[76], this is a microstratified brownish-yellow deposit of loose calcareous silt sands with few larger clasts.
Unit Xb (eight samples): similar to Xa, also microstratified but darker (brown) and finer-grained (sandy silts). Seven samples from this unit (ES1-7) were collected from a microstratified combustion structure (H44) at the top of this layer that yielded human fecal biomarkers[25]. The other sample was collected from underlying sediment.
Unit XI (one sample): this is a layer of loose brown silty sand.

**Ancient DNA extraction**. All work was conducted at University of Oklahoma LMAMR ancient DNA laboratory according to the following protocols for coprolite-derived materials.

For DNA extraction, approximately 200 mg were subsampled from each sample material and incubated on a rotator with 400 μl of 0.5 M EDTA and 100 μl of proteinase K (QIAGEN) for 4 h. After that, the samples were subjected to bead-beating with 750 μl of PowerBead solution (QIAGEN) and then extracted using the MinElute PCR Purification kit (QIAGEN) with a modified protocol (method B) described in Hagan et al.[77] and based on Dabney et al.[78], including two cleaning steps before final elution into two 30 μl of EB buffer (QIAGEN).

**Library preparation and sequencing**. Shotgun sequencing indexing libraries were constructed using the NEBNext DNA Library Prep Master Mix Set for 454 (New England Biolabs), following the "BEST" (Blunt-End-Single-Tube) method[79], with the hybridization of adapter oligos as per Meyer and Kircher[80]. Briefly, deaminated (C to U) bases were first partially removed (UDG-half) by uracil DNA glycosylase treatment using USER enzyme[81]. End overhangs were repaired, creating blunt-end phosphorylated regions for adapter ligation. Oligo adapters were ligated directly to blunt ends and filled in to create priming sites for index primers. After purification with a MinElute column (QIAGEN), indexed libraries were generated in triplicate for each sample using unique forward and reverse barcoded primers. See Supplementary Table 3 for adapter and oligo sequences. The triplicates were pooled, cleaned using Agencourt AMPure XP magnetic beads (Beckman Coulter), and then run on a Fragment Analyzer (Advanced Analytical) using the high sensitivity NGS standard protocol. Samples containing adapter dimers below the main peak for putative authentic endogenous DNA (i.e., 200–250 bp)[82], were further cleaned using AMPure XP magnetic beads in a PEG/NaCl buffer[83]. Cleaned samples were sequenced on Illumina NextSeq 500 platform (Illumina) at University of Bologna (Bologna, Italy), using paired-end $2 \times 75$ bp chemistry in order to obtain >1 Gbp of sequences per sample. Quality score exceeded Q30 for more than 95% of the sequenced bases. Sequencing data was pre-processed by retaining only merged reads matching the forward and reverse barcodes with no mismatches using AdapterRemoval[84].

**Bioinformatics analysis**. Sequences were analyzed using Burrows-Wheeler Aligner (BWA) aln algorithm and the entire set of bacterial and archaeal genomes available through NCBI RefSeq (downloaded on November 15th, 2017). In particular, we reduced the maximum accepted edit distance (i.e., the threshold of the maximum number of deletions, insertions, and substitutions needed to transform the reference sequence into the read sequence) to 1% (-n 0.01) and set the maximum number of gap opens (i.e., the threshold of the maximum number of gaps that can be initiated to match a given read to the reference) to 2, with long gap and seed length disabled (-e-1 -l 16500). These parameters are optimized for the specific types of errors generated by postmortem DNA damage during the alignment of ancient DNA to modern references, as indicated by Schubert and colleagues[85]. The aligned reads were further filtered for mapping quality >20, and only the hits with the best unique match (X0 = 1) were considered for analysis, in order to minimize the number of false positives. In order to retrieve the entire phylogeny of the assignment, database sequences were previously annotated with the "Tax" tags of the NCBI database using the reference-annotator tool of the MEGAN utils package[86]. We then used the calmd program of the samtools suite to recompute the MD tags (containing alignment information, such as mismatches) for all datasets.

To discriminate ancient DNA from modern-day contamination, we calculated the postmortem degradation score (PMDS) distributions[12]. Sequences with PMDS > 5 were considered ancient (over 5,000 years ago), as reported by Skoglund et al.[12], and used for further analysis. The outputs were transformed in sequence (.fasta) and annotation (.txt) files compatible with the QIIME command "make_otu_table.py", in order to create a table that contained the phylogenetic classification and the

abundance as number of reads for each specific taxon. This table was then collapsed at family, genus and species level using the command "summarize_taxa.py". The family-level relative abundance profiles of samples IX, Xa, ES1-7, Xb and XI were compared with publicly available data of the gut microbiota of human populations adhering to different subsistence strategies: urban Italians and Hadza hunter-gatherers from Tanzania (NCBI SRA, Bioproject ID PRJNA278393)[47], urban US residents, Matses hunter-gatherers and Tunapuco rural agriculturalists from Peru (NCBI SRA, BioProject ID PRJNA268964)[7]. Shotgun sequence datasets were downloaded and processed as El Salt samples, without applying the PMDS filter. 16S rRNA gene representative sequences of bacterial genera for which at least one species was present with more than 4 hits in one El Salt sample, were downloaded from the SILVA repository and used to build a phylogenetic tree by MUSCLE[87] and FastTree[88]. The tree was visualized using the GraPhlAn software[89]. Finally, bacterial species belonging to families that have recently been indicated as being common to the gut microbiome of hominids[38–45], were specifically sought and visualized for their abundance across El Salt samples by a heat map using the R software. Species membership in other source categories (i.e., human (or animal) oral and/or pathobiont, and environmental) was inferred by searching in PubMed the original article in which the taxonomy was first assigned to that organism, as well as more recent articles reporting its habitat description.

**Independent validation of taxonomic assignments**. To validate the taxonomic assignments of the metagenomic reads recovered from the El Salt samples, we used the same procedure adopted by Jensen and colleagues[30]. Specifically, we combined results from samples IX, Xa, ES1-7, Xb and XI (i.e., those positive for the presence of fecal biomarkers and/or associated with rich archaeological assemblages), then aligned the assigned reads to their respective reference genomes and examined edit distances, coverage distributions, and postmortem DNA damage patterns[14,53]. For the 24 bacterial families identified as common to hominid gut microbiome, we chose to further investigate bacterial species with ≥200 assigned reads (including strain-specific reads), for which at least 50 reads showed PMDS > 1 and at least one mismatch in the first 10 bases with respect to the reference genome. We then aligned the taxon-specific reads to the respective reference genome from the NCBI RefSeq database using bwa aln. MapDamage was used to estimate deamination rates (Supplementary Fig. 6)[90]. The breadth and depth of coverage were calculated with bedtools[91] and visualized with Circos[92] (Supplementary Fig. 7). Edit distances for all reads and filtered for PMDS > 1 were extracted from the bam files with the samtools view[93] and plotted in R (Supplementary Fig. 8). The negative difference proportion ($-\Delta$ %) was calculated considering the first 10 bases of reads with PMDS > 1. This metric was proposed by Hübler et al.[53] as a measure of decline in the edit distance distribution, with a $-\Delta$ % value of 1 indicating a declining distribution associated with an ancient DNA profile. Indeed, correct taxonomic assignments generally show a continuously declining edit distance distribution with only a few mismatches, mostly resulting from aDNA damage or divergence of the ancient genome from the modern reference. On the other hand, the mapping to an incorrect reference is associated with an increased number of mismatches, highlighted by the analysis of the edit distance distribution.

**mtDNA analysis and contamination estimate**. In order to detect human mtDNA, a similar procedure combining BWA (same parameters as above) and the reference-annotator tool of the MEGAN utils package, was applied to the entire set of mitochondrial sequences listed at the MitoSeqs website (https://www.mitomap.org/foswiki/bin/view/MITOMAP/MitoSeqs), including all the eukaryotic mitochondrial sequences available at the NCBI database. Only taxa detected in ancient sequences (i.e., with PMDS > 5) with more than 2 hits and not present in the control samples were retained. This procedure allowed us to detect ancient human traces beyond any reasonable doubt, eventually discarding more sequences than necessary.

In parallel, capture-enrichment for mtDNA sequencing was performed on the indexed libraries with a Neanderthal bait panel, as per the manufacturer's protocol (version 4.01, Arbor Biosciences). In short, libraries were denatured, blocked and incubated with baits for 48 h. After purification with streptavidin-coated magnetic beads, enriched libraries were amplified and concentrated, before being subjected to a second round of capture. Final libraries were sequenced on an Illumina NextSeq 500 platform (Illumina) at University of Bologna (Bologna, Italy), as described above. As for read processing, we used Schmutzi[94] to determine the endogenous consensus mtDNA sequence and to estimate present-day human contamination. Reads were mapped to the mt-Neanderthal reference sequence (NC_011137) and filtered for MAPQ ≥ 30. Haploid variants were called using the endoCaller program implemented in Schmutzi and only the variants with a posterior probability exceeding 50 on the PHRED scale (probability of error: 1/100,000) were retained for further analysis. The PMDS profile of the reads was computed by PMDtools[12]. The negative difference proportion ($-\Delta$ %) was calculated using only reads with PMDS > 1[53]. Contamination estimates were obtained using Schmutzi's mtCont program and a database of putative modern contaminant mtDNA sequences. Samples with >1,000 PMDS > 1 reads, breadth of coverage >10%, $-\Delta$ % ≥ 0.9 and mtCont contamination less than 2% were considered to contain ancient human mtDNA.

**Statistics and reproducibility**. No replicates are included, all samples herein analyzed are unique.

Wilcoxon test was used to assess differences between samples IX, Xa, ES1-7, Xb and XI (i.e., those positive for the presence of fecal biomarkers and/or associated with rich archaeological assemblages), and samples from SU V (i.e., with no or very few archeological remains) in the number of PMDS > 5 reads, as well as in the relative abundances of the 24 families common to the gut microbiome of hominids[38–45].

The significance of data separation in the Bray-Curtis-based Principal Coordinates Analysis between the family-level relative abundance profiles of samples IX, Xa, ES1-7, Xb and XI, and the gut microbiota of urban Italians and Hadza hunter-gatherers from Tanzania[47], urban US residents, Matses hunter-gatherers and Tunapuco rural agriculturalists from Peru[7] was tested using a permutation test with pseudo-F ratio.

**Reporting summary**. Further information on research design is available in the Nature Research Reporting Summary linked to this article.

## Data availability

Sequencing data are accessible at the European Nucleotide Archive (ENA; project ID PRJEB41665). Source data are available as Supplementary Data. All sediment samples are readily available from the authors, subject to exhaustion.

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

## Acknowledgements
We thank F. D'Amico (Department of Pharmacy and Biotechnology, University of Bologna, Bologna, Italy) and E. Cilli (Department of Cultural Heritage, University of Bologna, Ravenna, Italy) for their valuable help in library preparation. This research was supported by the US National Institutes of Health, grant number R01GM089886 (C.W. and C.L.). S.B. was supported by the European Research Council (ERC) under the European Union's Horizon 2020 research and innovation programme (grant agreement No 724046 - SUCCESS). Archaeological research at El Salt is funded by Spanish I+D Project HAR2008-06117/HIST (C.M., C.H. and B.G.).

## Author contributions
M.C., S.T., S.R., S.L.S. and C.W.: conceptualization; C.M., C.H., B.G. and A.S.: field work, excavation, and sampling; S.L.S., C.A.H. and S.T.: DNA extraction and library preparation; A.A.: sequencing; S.R.: bioinformatics analysis; P.B., M.C., C.L., C.W. and S.B.: resources; M.C. and S.L.S.: supervision; M.C., S.T. and S.R.: writing—original draft; C.W., C.L., S.B., C.M., A.S., E.B., A.A., C.A.H. and S.L.S.: writing—review & editing. All authors gave final approval for publication.

## Competing interests
The authors declare no competing interests.
