## [Peer Review File · Communications Biology]

Reviewers' comments:

Reviewer #1 (Remarks to the Author):

The authors characterized the metagenomes of potential Neanderthal samples that may be the oldest so far. Sample collection, and the appropriate ancient DNA techniques and analyses were applied.

Major comments:

The very low amount of DNA in the samples is surprising given that these are fecal contents. Was this the reason the authors showed data in terms of numbers of hits? Can the authors provide results in terms of relative abundances?

The authors also make suggestions of the bacteria present in the samples being SCFA producers based on information of modern subjects' microbiomes. It should be noted that modern individuals (westernized or isolated cultures) have, for the most part, different diets compared to the Neanderthals' diet; thus, it may be reasonable that the substrate for SFCA producers differ between the Neanderthal and modern individuals. To conclusively show the presence of SFCA producers with the sequence information analyzed, the authors need to provide the predicted functional profiles to find the presence (and relative abundance) of pathways associated with SCFA production. At the very least, the authors can provide functional level predictions at lower levels of classification (i.e. amino acid metabolism, lipid biosynthesis, etc).

Importantly, why was not a comparator group(s) included? It would be important to compare the data to other publicly available metagenomic datasets of ancient and modern origin to have a reference point.

Line 76: The authors mention that the samples are of a human origin based on the lipid composition of the samples. Can they provide the number of reads matching human sequences? Can they determine the number of reads matching Neanderthal sequences? This would be further confirmation of the origin of the sample.

Minor comments:

Abstract:

Line 37: Please modify 'bacterial species' to 'bacterial genera', as authors mention a list of bacterial genera in Line 40. Alternatively, please provide the bacterial taxonomic classification at the species level.

Main text:

Lines 92-94: Is there a threshold sequence identity used to compare bacterial sequences from the Neanderthal samples to the reference genomes?

Lines 102-105: Were these bacterial families also identified in the blank samples? Please confirm.

Lines 164-169: Can the authors include the relative abundances for each bacterial genus?

Lines 180-185: Can the authors include the relative abundances for each bacterial species?

Lines 268-271: Can the authors expand on the criteria for hits selection? For example, what was the criteria for the mapping analysis? Was there a threshold? How did the authors minimize the number of false positives?

Reviewer #2 (Remarks to the Author):

The main question with this study is: Which signals are real and which ones are contaminants. Personally, I can't be sure yet about everything but I'd like to be convinced more.

Table S1: Looks like some serious filtering was done. My main worry is however that when you look at the ratio of ancient bacterial reads and high quality joined reads between controls and samples that you get about the same ratio. The samples don't really appear to have many more ancient reads, relatively. Clearly, several species typically associated with reagent contamination are also classified as being ancient (Table S2). This leads me to fear that many of the other ancient reads might also not be ancient. Do you perhaps need to be more strict? What happens if you set different thresholds? Are perhaps additional analyses needed? I'd perhaps like to see some kind of additional analyses on some of the most abundant signals.

If you look at the DNA damage levels of different bacterial groups, do you notice differences between for example Spirochaetaceae, Bifidobacteriaceae and several of the proteobacterial and Bacteroidetes groups?

I'm not that convinced (yet) that the method for determining which reads are ancient and which reads are not is really was sufficient. Skoglund et al. were looking at human/neanderthal DNA. That seems like child's play compared to the complexity of these kinds of samples; this pipeline might not yet be sufficient to really get the picture as clear as possible.

When analyzing the data from Figures 3, 4 and Table S3 I do however see several signals of interest. While many of the signals are also found in modern westernized humans the main thing makes me a bit more confident that we're dealing with at least some real signals in ES1-ES7 is that you also find a lot of Spirochaetaceae (*Treponema*), *Prevotella* and *Desulfovibrio*. Spirochaetaceae are typically no longer found in westernized humans; they are still found quite a bit in hunter/gatherer populations. The Spirochaetaceae finding thus makes perfect sense. The abstract however focuses on signals which do not really show that a real signal was found. (Some of) These "modern" bugs might (theoretically) also be due to contamination. I'd focus more on what you'd expect in hunter/gatherers perhaps.

What also seems to indicate that they are at least describing quite a lot of real ancient signals is that large differences are found between ES1-ES7 and the other non-negative control samples. Maybe explicitly mention this in abstract? "are abundantly represented "only" across unit X samples"

I personally btw do not hold too much stock in the ability of this analysis to always go down to the species level for all species; this is mainly just possible using samples from western humans for which reference libraries are quite developed. Also: DNA damage. Several of the references are thus probably irrelevant/misleading (Table S3).

Treponema maltophilum, *Treponema 184 medium*, *Treponema socranskii* and *Treponema vincentii*), which have been associated with modern 185 oral and dental diseases in humans (56-65).

I'd strongly recommend reinterpreting your *Treponema* findings.

199 presence of known SCFA producers, such as *Blautia*, *Dorea*, *Roseburia*, *Rumunicoccus*,
200 *Subdoligranulum* and *Faecalibacterium*, as well as *Bifidobacterium*,

Bifidobacteria are also "well known" SCFA producers (acetate, lactate). *B. adolescentis* is part of many complex syntrophic networks. Replace ", as well as" with "and" perhaps?

I think it is important that still a bit more attention is given to the possibility of contamination in your data in the manuscript and why several/most of your (relevant) signals are not

contamination.

"In the current scenario where we are witnessing a wholesale
206 loss of bacterial diversity in the GM of the cultural "west", with the parallel rise in dysbiosis-
207 autoimmune and inflammatory disorders (71), the identification of these evolutionarily
integral taxa 208 of the human holobiont may benefit practical applications favoring their retention
among 209 populations living in or transitioning to increasingly microbially deplete contexts."

I very strongly agree with this part.

Minor points:

In the abstract I'd replace "probiotic" with "commensal" or "beneficial commensal".

"Studies aimed at exploring the ancestral traits of the human GM are therefore encouraged,
50 as a unique evolutionary perspective to improve our knowledge of GM assembly and
interactions 51 with the human host, a key prerequisite for discerning the fundamental traits for
our health (2)."

Sentence is too long or simply needs revising.

69 M components by shotgun metagenomics
70 analysis

Remove "s" from metagenomics?

In the latter subgroup, we can count
141 several species from Lachnospiraceae (including well-known short-chain fatty acid (SCFA)
142 producers of modern human guts, such as Blautia, Coprococcus, Dorea, Fusicatenibacter and
143 Roseburia spp.) and Ruminococcaeae families

Many bacteria produce various kinds of short-chain fatty acids. Perhaps just say
"well-known (beneficial) commensal inhabitants of modern human guts, such as..."

It would have been good if the actual numbers would have also been provided (for each sample
individually), like they were in Table S2 (for the controls), also in order to better compare the
controls and the samples in regards to their composition (see previous point). Figures 3 and 4
already show quite a bit of detail, but I'd like to have the numbers in a simple excel sheet
(supplemental) or a supplemental table.

due to its very
146 promising potential as a biomarker of healthy GM (46).

"a" healthy GM?

This ancient human GM core may inform on
198 evolutionary symbioses with a strong potential to be most impactful on our health

Would recommend revising sentence. (may inform on?)

Best regards,
Marcus de Goffau

Reviewer #3 (Remarks to the Author):

Rampelli et al apply metagenomic techniques to sediment samples associated with past Neanderthal occupation in an attempt to identify hominid-associated gut microorganisms. While an interesting idea, I am not convinced that the evidence is strong enough to conclusively say that the microbes detected were of Homo gut origin.

An equally plausible interpretation is that the microbial taxa detected were from faeces deposited by other animals, either contemporaneously or throughout the 50 ky history of the site (cells/DNA could have penetrated from above). The microbial taxa listed (lines 122-127 and 141-159) are also commonly found in the guts of many other mammalian species, and in my opinion, not enough has been done to demonstrate that they are of Homo origin.

For one, species identification (especially from ancient DNA) can be difficult if the proper reference genomes are not present -- reads can be assigned sub optimally or erroneously due to database bias (see Figure 4 from Warinner et al. 2017: A Robust Framework for Microbial Archaeology. *Annu. Rev. Genom. Hum. Genet.* <https://doi.org/10.1146/annurev-genom-091416-035526>). The veracity of the microbial species assignments by the authors must be tested using edit-distances (see Hubler et al. 2019: HOPS: automated detection and authentication of pathogen DNA in archaeological remains. *Genome Biology*. DOI: 10.1186/s13059-019-1903-0).

Overall, I think that the authors propose an interesting idea, but more work is needed for it to be robust and convincing.

Minor comments:

Lines 267-268: "entire set of bacterial genomes of the NCBI database (downloaded on November 15th, 2017)"

The authors may wish to be a bit more specific here, as there are multiple sets of bacterial genomes available at the NCBI, e.g. RefSeq, GenBank, etc. Also, I presume that their selection includes Archaea?

Figure S1, which depicts DNA deamination plots is a bit confusing to me for a couple of reasons.

1. Typically, such plots depict the alignment of a sample's reads to a single reference genome. How is this done for the metagenomic samples in this paper, are they summed across all microbial species?

2. The methods state that the authors used a UDG-half treatment, but the plots in this figure do not look like they have had this treatment. UDG-half treatment should drastically reduce the deamination signal observable to the last few terminal bases. (see figure 2 from (Rohland et al. 2015: Partial uracil-DNA-glycosylase treatment for screening of ancient DNA. *Phil. Trans. R. Soc. B* 370:20130624 <http://doi.org/10.1098/rstb.2013.0624>)

Referee expertise:

Referee #1: aDNA microbiome

Referee #2: Metagenomics

Referee #3: aDNA, microbiomes

Reviewers' comments:

Reviewer #1 (Remarks to the Author):

The authors characterized the metagenomes of potential Neanderthal samples that may be the oldest so far. Sample collection, and the appropriate ancient DNA techniques and analyses were applied.

Major comments:

The very low amount of DNA in the samples is surprising given that these are fecal contents. Was this the reason the authors showed data in terms of numbers of hits? Can the authors provide results in terms of relative abundances?

The Reviewer is right, the amount of DNA we recovered was relatively low, but we must take into consideration that we were working with sedimentary samples dating back to 60.7-45.2 kya, containing millimetric phosphatic coprolites. In other words, the samples contained only traces of fecal material, which could explain the low amount of extracted DNA. As the Reviewer suggested, we calculated the relative abundance of members belonging to the 24 families common to the gut microbiome of hominids, in all samples positive for the presence of fecal biomarkers and/or associated with rich archaeological assemblages, i.e., in samples IX, Xa, ES1-7, Xb and XI (please see the figure below). While these data appear promising, we are aware that they probably do not reflect, if not partially, the proportions of the EI Salt fecal samples at the time of deposition. Indeed, due to the extremely old age of the samples and their sedimentary nature, it was more than reasonable to expect a relevant degree of degradation of the microbial DNA, which might be different for the various microbiome components. In light of all these considerations, if the Reviewer agrees, we would prefer not to include this information in our work, limiting our findings to the distribution of the number of hits.

Figure 1. Bar plot showing the relative abundance of the 24 families common to the gut microbiome of hominids, in the EI Salt site. All sedimentary samples positive for the presence of fecal biomarkers (Sistiaga et al., 2014) and/or associated with rich archaeological assemblages, i.e., samples IX, Xa, ES1-7, Xb and XI, were included.

The authors also make suggestions of the bacteria present in the samples being SCFA producers based on information of modern subjects' microbiomes. It should be noted that modern individuals (westernized or isolated cultures) have, for the most part, different diets compared to the Neanderthals' diet; thus, it may be reasonable that the substrate for SCFA producers differ between the Neanderthal and modern individuals. To conclusively show the presence of SCFA producers with the sequence information analyzed, the authors need to provide the predicted functional profiles to find the presence (and relative abundance) of pathways associated with SCFA production. At the very least, the authors can provide functional level predictions at lower levels of classification (i.e. amino acid metabolism, lipid biosynthesis, etc).

We fully understand and share this point, but in our work we never refer to the actual SCFA production, on the contrary we limit to reporting the presence of SCFA producers of the gut microbiome (particularly in samples ES1-7). Indeed, to the best of our knowledge, microorganisms such as *Blautia*, *Coprococcus*, *Dorea*, *Roseburia*, *Faecalibacterium* and others - all detected in ES1-7 samples - are universally recognized as SCFA producers, due to their inherent fermentation properties (e.g. Flint et al., 2012 doi: 10.4161/gmic.19897; Flint et al., 2015 doi: 10.1017/S0029665114001463; Reichardt et al., 2018 doi: 10.1038/ismej.2017.196). Consistently, in the text we always refer to their presence, without ever mentioning the SCFA production.

We thank the Reviewer for the suggestion to extend our findings at the functional level but unfortunately, due to the relevant degree of fragmentation of the recovered aDNA, the functional prediction from our ancient reads was not reliable.

Importantly, why was not a comparator group(s) included? It would be important to compare the data to other publicly available metagenomic datasets of ancient and modern origin to have a reference point.

We totally agree with the Reviewer, such a comparison would be highly informative indeed. However, as discussed in the first point, we believe that the compositional structure of our samples in terms of relative abundance can be affected by some biases, due to the old age and sedimentary nature of the samples, potentially resulting in differential rates of DNA degradation between the different microbiome components (please see above for discussion of this point).

As suggested by her/him, we performed a comprehensive gut microbiome meta-analysis, including publicly available data from different human populations - Hadza hunter-gatherers (Tanzania), Matses hunter-gatherers (Peru), Tunapuco rural agriculturalists (Peru), urban Italian and US residents - and the El Salt samples (those positive for the presence of fecal biomarkers and/or associated with rich archaeological assemblages, i.e., samples IX, Xa, ES1-7, Xb and XI). As shown in the figure below, although the El Salt samples cluster apart in the bidimensional Bray-Curtis-based PCoA plot, along PCo1 (accounting for 27% of the total variance) they are closer to Tunapuco and Matses, supporting the “ancestral configuration” of the gut microbial communities of rural agriculturalists and hunter-gatherers (Obregon-Tito et al., 2015 doi: 10.1038/ncomms7505). However, for the reasons mentioned above, we would prefer not to include this analysis in the manuscript.

Figure 2. Principal Coordinates Analysis based on Bray-Curtis distances between the family-level relative abundance profiles of El Salt samples and the gut microbiota of human populations adhering to different subsistence strategies. The 24 families common to the gut microbiome of hominids were considered (see also Figure 1). For the El Salt site, samples IX, Xa, ES1-7, Xb and XI were included, i.e., those positive for the presence of fecal biomarkers (Sistiaga et al., 2014) and/or associated with rich archaeological assemblages. Publicly available data from the following human populations were included: urban Italians and Hadza hunter-gatherers from Tanzania (Schnorr et al., 2014 doi: 10.1038/ncomms4654), urban US residents, Matses hunter-gatherers and Tunapuco rural agriculturalists from Peru (Obregon-Tito et al., 2015 doi: 10.1038/ncomms7505). PCo1 and PCo2 account for 27% and 20% of the total variance, respectively. A significant separation between groups was found ($p=0.0001$, permutation test with pseudo-F ratio).

Line 76: The authors mention that the samples are of a human origin based on the lipid composition of the samples. Can they provide the number of reads matching human sequences? Can they determine the number of reads matching Neanderthal sequences? This would be further confirmation of the origin of the sample.

We are grateful to the Reviewer for this valuable suggestion.

In the revised version of our manuscript, we searched for human mitochondrial DNA sequences in the PMDS-filtered metagenomes from the whole sample set. According to our findings, ancient human mtDNA was detected in nearly all ES1-7 samples from SU X (please see the new Figure 2B of the main text). Furthermore, to strengthen our findings, we performed a new sequencing totally dedicated to mtDNA, enriched by hybridization capture with a Neanderthal bait panel (Arbor Biosciences). The results are provided in the new Figure 2B of the main text and Table S3. Specifically, we used Schmutzi (Jensen et al., 2019 doi: 10.1038/s41467-019-13549-9) to determine the endogenous consensus mtDNA sequence and to estimate present-day human contamination. Only variants with a posterior probability exceeding 50 on the PHRED scale (probability of error: 1/100,000) and estimation of modern contamination less than 2% were retained. Results were also filtered for >1,000 reads with PMDS >1, breadth of coverage >10%, $-\Delta \% \geq 0.9$. Based on this new analysis, samples ES1, ES2, ES5 and Xb showed ancient human mtDNA traces, which strengthens their human origin. These new data are discussed in L107-116, and the new methods are described in L354-379.

Minor comments:

Abstract:

Line 37: Please modify 'bacterial species' to 'bacterial genera', as authors mention a list of bacterial genera in Line 40. Alternatively, please provide the bacterial taxonomic classification at the species level.

We apologize for the inaccuracy. The change has been made (L37).

Main text:

Lines 92-94: Is there a threshold sequence identity used to compare bacterial sequences from the Neanderthal samples to the reference genomes?

We apologize for the lack of details.

For the mapping analysis, we used BWA.aln, a fast, light-weight tool that aligns relatively short nucleotide sequences against long reference sequences such as bacterial genomes. We used this tool with the same parameters used by Skoglund and colleagues to separate endogenous ancient DNA from modern-day contamination in a Siberian Neanderthal (Skoglund et al., 2014 doi: 10.1073/pnas.1318934111). In particular, we reduced the maximum accepted edit distance (i.e., the threshold of the maximum number of deletions, insertions, and substitutions needed to transform the reference sequence into the read sequence) to 1% (-n 0.01) and set the maximum number of gap opens (i.e., the threshold of the maximum number of gaps that can be initiated to match a given read to the reference) to 2, with long gap and seed length disabled (-e-1 -l 16500). These parameters are optimized for the specific types of errors generated by post-mortem DNA damage during the alignment of ancient DNA to modern references, as indicated by Schubert and colleagues (Schubert et al., 2012 doi: 10.1186/1471-2164-13-178). The aligned reads were further filtered for mapping quality >20, and only the hits with the best unique match (X0=1) were considered for analysis in order to minimize the number of false positives. Please see L299-310.

Lines 102-105: Were these bacterial families also identified in the blank samples? Please confirm.

As shown in Table S2, species belonging to the families listed in those lines (i.e., *Streptomycetaceae*, *Pseudonocardiaceae*, *Micromonosporaceae*, *Nocardiaceae*, *Mycobacteriaceae*, *Microbacteriaceae* and *Nocardioidaceae*), have also been detected in some blanks, mainly in the extraction blank. However, it should be noted that the hit count was mostly 1 and reached a maximum of 3. In other words, we can say that there is essentially no overlap with the sample dataset.

Lines 164-169: Can the authors include the relative abundances for each bacterial genus?

Lines 180-185: Can the authors include the relative abundances for each bacterial species?

As discussed above, if the Reviewer agrees, we would prefer not to include this information but to limit the presentation of the results to the distribution of the number of hits as shown in Figure 4.

Lines 268-271: Can the authors expand on the criteria for hits selection? For example, what was the criteria for the mapping analysis? Was there a threshold? How did the authors minimize the number of false positives?

Please, see the answer to the second minor comment.

The methods section has been modified accordingly (L299-310).

Reviewer #2 (Remarks to the Author):

The main question with this study is: Which signals are real and which ones are contaminants. Personally, I can't be sure yet about everything but I'd like to be convinced more.

We understand this general concern and in the revised version of our manuscript, we have done our best to further strengthen our findings, also taking into consideration the valuable suggestions of the other Reviewers. New experiments were performed to detect human mtDNA traces in the whole sample set, and the microbial DNA sequences were re-analyzed in depth following the approach recently used by Jensen et al., 2019 (doi: 10.1038/s41467-019-13549-9). Please, see below for all new data and analysis.

Table S1: Looks like some serious filtering was done. My main worry is however that when you look at the ratio of ancient bacterial reads and high quality joined reads between controls and samples that you get about the same ratio. The samples don't really appear to have many more ancient reads, relatively. Clearly, several species typically

associated with reagent contamination are also classified as being ancient (Table S2). This leads me to fear that many of the other ancient reads might also not be ancient. Do you perhaps need to be more strict? What happens if you set different thresholds? Are perhaps additional analyses needed? I'd perhaps like to see some kind of additional analyses on some of the most abundant signals.

We are grateful to the Reviewer for these valuable suggestions.

In this revised version of the manuscript, we have performed a new analysis comparing the fraction of reads with PMDS >5 per million reads between samples IX, Xa, ES1-7, Xb and XI (i.e. those positive for the presence of fecal biomarkers and/or associated with rich archaeological assemblages), and samples from SU V (i.e., negative samples, with no or very few archeological remains). According to our findings, the samples containing traces of microscopic coprolites and fecal lipid biomarkers of presumed archaic human origin showed a greater abundance of PMDS >5 reads, possibly as a result of the presence of human fecal sediment. The results of this new analysis are provided as Figure S2 and discussed in the main text (L101-106).

As for the classification of species typically associated with reagent contamination as ancient, it must be said that most contaminants of laboratory reagents, DNA extraction kits and other laboratory equipment include environmental microorganisms, such as water-borne bacterial genera (e.g. *Pseudomonas*, *Stenotrophomonas*, *Xanthomonas*, *Ralstonia* and *Bacillus*), and soil and plant-associated bacteria (e.g. *Sphingobacteriaceae*, *Bradyrhizobiaceae*, *Methylobacterium* and *Phyllobacteriaceae*) (Olomu et al., 2020 doi: 10.1186/s12866-020-01839-y). In light of the sedimentary nature of our samples, it is not surprising that such microorganisms may not only be present but also actually be ancient. Looking at Table S2, it is evident that some of these microorganisms were detected in blanks (mainly in the extraction blank) but with a hit count of mostly 1 to a maximum of 4, thus suggesting minimal interference from what has recently been called the “kitome” (i.e., contaminants in DNA extraction kits and reagents).

Finally, as suggested by the Reviewer, in this revised version of the manuscript, we have provide a new analysis of the reads by applying the approach recently used by Jensen et al., 2019 (doi: 10.1038/s41467-019-13549-9). In particular, all the reads retrieved from the samples IX, Xa, ES1-7, Xb and XI (i.e., those positive for the presence of fecal biomarkers and/or associated with rich archaeological assemblages) were first annotated and, subsequently, the ancient origin of each taxon was authenticated by computing 3 parameters: fraction of reads with PMDS >1, and $-\Delta$ % and 5' deamination rate of PMDS >1 reads. Data from this new analysis are provided in the new Table 1 and as Supplementary Figures S3-S5 (MapDamage plots, coverage plots and edit distance distribution). Based on these new results, we were able to confirm the presence of several species belonging to the GM families of hominids, showing a read profile consistent with their ancient origin, as discussed in L177-187. Please, see L331-352 for new methods.

If you look at the DNA damage levels of different bacterial groups, do you notice differences between for example Spirochaetaceae, Bifidobacteriaceae and several of the proteobacterial and Bacteroidetes groups?

As requested, please find below the MapDamage plots for Bacteroidetes, Proteobacteria, *Spirochaetaceae* and *Bifidobacteriaceae*, which show overall comparable profiles. Indeed, our fecal sediments were dispersed in a soil matrix, containing not only modern bacterial contaminants, but also ancient environmental bacteria that likely grew around or stayed close to the micro-coprolites and were therefore detected by our analysis.

Figure 3. MapDamage plots for Bacteroidetes, Proteobacteria, *Spirochaetaceae* and *Bifidobacteriaceae*. Only bacterial reads with PMDS >1 from the El Salt samples IX, Xa, ES1-7, Xb and XI were considered.

I'm not that convinced (yet) that the method for determining which reads are ancient and which reads are not is really was sufficient. Skoglund et al. were looking at human/neanderthal DNA. That seems like child's play compared to the

complexity of these kinds of samples; this pipeline might not yet be sufficient to really get the picture as clear as possible.

We understand the Reviewer's concern.

Skoglund et al. actually used that pipeline in a work focused on human/Neanderthal DNA. They looked at cytosine deamination patterns and defined a postmortem degradation score (PMDS) to identify degraded DNA sequences that are unlikely to originate from modern contamination. Specifically, positive values of PMDS indicate support for the sequence being genuinely ancient. The same tool was subsequently applied to bacterial sequences in several contexts (e.g. Jensen et al., 2019 doi: 10.1038/s41467-019-13549-9; Borry et al., 2020 doi: 10.7717/peerj.9001).

Anyway, in an attempt to dispel the doubts related to the method used, as anticipated above, in this revised version of the manuscript we have provided a new analysis of the reads using the same approach by Jensen et al., 2019 (doi: 10.1038/s41467-019-13549-9). Please, see also the reply to the previous point. The results of this new analysis are provided in the new Table 1 and Supplementary Figures S3-S5, and described in the main text in L177-187. This analysis has allowed us to confirm the ancient origin for several species, members of the GM families of hominids, detected in the El Salt samples.

When analyzing the data from Figures 3, 4 and Table S3 I do however see several signals of interest. While many of the signals are also found in modern westernized humans the main thing makes me a bit more confident that we're dealing with at least some real signals in ES1-ES7 is that you also find a lot of Spirochaetaceae (*Treponema*), *Prevotella* and *Desulfovibrio*. Spirochaetaceae are typically no longer found in westernized humans; they are still found quite a bit in hunter/gatherer populations. The Spirochaetaceae finding thus makes perfect sense. The abstract however focuses on signals which do not really show that a real signal was found. (Some of) These "modern" bugs might (theoretically) also be due to contamination. I'd focus more on what you'd expect in hunter/gatherers perhaps.

We are grateful to the Reviewer for raising this relevant point.

In the revised version of our manuscript, we do not fail to mention the presence of these "old friends" – as Prof. Blaser defined them – as putative members of the Neanderthal GM and therefore as evolutionary taxa likely to be integral to human biology (L235-238).

What also seems to indicate that they are at least describing quite a lot of real ancient signals is that large differences are found between ES1-ES7 and the other non-negative control samples. Maybe explicitly mention this in abstract? "are abundantly represented "only" across unit X samples"

We thank the Reviewer for this suggestion.

The text has been changed accordingly (L39).

I personally btw do not hold too much stock in the ability of this analysis to always go down to the species level for all species; this is mainly just possible using samples from western humans for which reference libraries are quite developed. Also: DNA damage. Several of the references are thus probably irrelevant/misleading (Table S3).

We understand the Reviewer's concern and share this point.

The read sequences were compared with available references, which mostly include species from western human subjects. Therefore, the species-level identification may actually be partially biased. On the other hand, for the mapping analysis, we used BWA.aln with the same parameters used by Skoglund and colleagues to separate endogenous ancient DNA from modern-day contamination in a Siberian Neanderthal (Skoglund et al., 2014 doi: 10.1073/pnas.1318934111).

In particular, we reduced the maximum accepted edit distance (i.e., the threshold of the maximum number of deletions, insertions, and substitutions needed to transform the reference sequence into the read sequence) to 1% (-n 0.01) and set the maximum number of gap opens (i.e., the threshold of the maximum number of gaps that can be initiated to match a given read to the reference) to 2, with long gap and seed length disabled (-e-1 -l 16500). These parameters are optimized for the specific types of errors generated by post-mortem DNA damage during the alignment of ancient DNA to modern references, as indicated by Schubert and colleagues (Schubert et al., 2012 doi: 10.1186/1471-2164-13-178). The aligned reads were further filtered for mapping quality >20, and only the hits with the best unique match (X0=1) were considered for analysis in order to minimize the number of false positives. No less important, as mentioned above, the results were further verified using the same approach by Jensen et al., 2019 (doi: 10.1038/s41467-019-13549-9).

Treponema maltophilum, *Treponema 184 medium*, *Treponema socranskii* and *Treponema vincentii*, which have been associated with modern 185 oral and dental diseases in humans (56-65).

I'd strongly recommend reinterpreting your *Treponema* findings.

We thank the Reviewer for her/his comment.

The *Treponema* species listed in brackets (i.e., *T. maltophilum*, *T. medium*, *T. socranskii* and *T. vincentii*) have been variously associated with oral and dental diseases (periodontal lesions, plaque, etc.), as repeatedly reported in the literature (e.g. Takeuchi et al., 2001 doi: 10.1902/jop.2001.72.10.1354; Asai et al., 2002 doi: 10.1128/JCM.40.9.3334-3340.2002; Edwards et al., 2003 doi: 10.1046/j.1365-2672.2003.01901.x; Lee et al., 2005 doi: 10.1128/IAI.73.1.268-

276.2005). We are totally aware that the *Treponema* genus also includes species considered “old friends” (and disappeared in the Western microbiomes) but, to the best of our knowledge, these are different species, such as *T. porcinum*, *T. bryantii*, *T. succinifaciens*, *T. parvum* and *T. berlinense*, as we have identified in the GM of the Hadza hunter-gatherers (Soverini et al., 2016 doi: 10.3389/fmicb.2016.01058).

In light of these considerations, if the Reviewer agrees, we would prefer to leave this sentence as it is but we have nevertheless specified in the text that *Treponema* and others are considered “old friends” (L235-238 and please, see also the reply to a previous point).

199 presence of known SCFA producers, such as *Blautia*, *Dorea*, *Roseburia*, *Rumunicoccus*,
200 *Subdoligranulum* and *Faecalibacterium*, as well as *Bifidobacterium*,

Bifidobacteria are also “well known” SCFA producers (acetate, lactate). *B. adolescentis* is part of many complex syntrophic networks. Replace “, as well as” with “and” perhaps?

Sorry for the oversight.

Bifidobacteria are actually acetate producers so, as suggested, we have replaced “as well as” with “and” (L230).

I think it is important that still a bit more attention is given to the possibility of contamination in your data in the manuscript and why several/most of your (relevant) signals are not contamination.

We thank the Reviewer for this suggestion.

In L98-101, we wrote: “The same procedure was applied to extraction, library and PCR blanks, resulting in the retrieval of a minimal number of 144, 1, and 42 ancient bacterial sequences, respectively. Ancient reads from blanks were assigned to 116 bacterial species that showed no overlap with the sample dataset (table S2).” This is essentially the approach we have applied to exclude – as best as possible – the signals due to samples contamination with modern environmental DNA during DNA extraction and library preparation. Furthermore, all the conclusions about the putative Neanderthal GM that we drew in our work were based on reads filtered for PMDS >5 and, in the revised version of the manuscript, we performed new analysis following the approach used by Jensen et al. (2019 doi: 10.1038/s41467-019-13549-9), to further support the ancient nature of our data. Nonetheless, we are aware that the risk of modern DNA contamination can never be 100% excluded. Therefore, as suggested by the Reviewer, in the new version of our manuscript, we have included some warnings about the possible risk of modern DNA contamination (L225-227).

“In the current scenario where we are witnessing a wholesale
206 loss of bacterial diversity in the GM of the cultural “west”, with the parallel rise in dysbiosis-related 207 autoimmune
and inflammatory disorders (71), the identification of these evolutionarily integral taxa 208 of the human holobiont may
benefit practical applications favoring their retention among 209 populations living in or transitioning to increasingly
microbially deplete contexts.”

I very strongly agree with this part.

We are delighted and honored that the Reviewer strongly agrees with our conclusions.

Minor points:

In the abstract I'd replace “probiotic” with “commensal” or “beneficial commensal”.

The wording “probiotic gut components” has been replaced with “beneficial gut commensals” (L39).

“Studies aimed at exploring the ancestral traits of the human GM are therefore encouraged,
50 as a unique evolutionary perspective to improve our knowledge of GM assembly and interactions 51 with the human
host, a key prerequisite for discerning the fundamental traits for our health (2).”

Sentence is too long or simply needs revising.

The indicated sentence has been shortened (L50-52).

69 M components by shotgun metagenomics
70 analysis

Remove “s” from metagenomics?

The “s” has been removed (L70).

In the latter subgroup, we can count
141 several species from Lachnospiraceae (including well-known short-chain fatty acid (SCFA)
142 producers of modern human guts, such as *Blautia*, *Coprococcus*, *Dorea*, *Fusicatenibacter* and
143 *Roseburia* spp.) and Ruminococcaeae families

Many bacteria produce various kinds of short-chain fatty acids. Perhaps just say
"well-known (beneficial) commensal inhabitants of modern human guts, such as..."

Thanks for the correction. The text has been changed as suggested (L158).

It would have been good if the actual numbers would have also been provided (for each sample individually), like they were in Table S2 (for the controls), also in order to better compare the controls and the samples in regards to their composition (see previous point). Figures 3 and 4 already show quite a bit of detail, but I'd like to have the numbers in a simple excel sheet (supplemental) or a supplemental table.

Following the Reviewer's suggestion, we prepared a table with the relative abundances of the bacterial families detected in all the samples. We have chosen this phylogenetic level to reduce the size of the table. However, as there are more than 400 entries, if the Reviewer agrees, we would prefer not to include it in the supplementary material.

due to its very
146 promising potential as a biomarker of healthy GM (46).

"a" healthy GM?

An "a" has been added (L163).

This ancient human GM core may inform on
198 evolutionary symbioses with a strong potential to be most impactful on our health

Would recommend revising sentence. (may inform on?)

The sentence has been revised (L227-228).

Reviewer #3 (Remarks to the Author):

Rampelli et al apply metagenomic techniques to sediment samples associated with past Neanderthal occupation in an attempt to identify hominid-associated gut microorganisms. While an interesting idea, I am not convinced that the evidence is strong enough to conclusively say that the microbes detected were of Homo gut origin.

We are honored that the Reviewer finds our work of some interest.
As also suggested by the other Reviewers, in the revised version of our manuscript, much additional work has been done to prove the *Homo* origin of the samples and to refine the analysis of the ancient reeds. Please, see below for any details.

An equally plausible interpretation is that the microbial taxa detected were from faeces deposited by other animals, either contemporaneously or throughout the 50 ky history of the site (cells/DNA could have penetrated from above). The microbial taxa listed (lines 122-127 and 141-159) are also commonly found in the guts of many other mammalian species, and in my opinion, not enough has been done to demonstrate that they are of Homo origin.

In order to consolidate the *Homo* origin of the fecal remains found in the EI Salt samples, we searched for human mitochondrial DNA (mtDNA) sequences in PMDS-filtered metagenomes from the 14 EI Salt archeological sedimentary samples. According to our findings, ancient human mtDNA was detected in nearly all ES1-7 samples from SU X, while no traces of animal DNA were detected in these and other samples. Moreover, to strengthen our findings, we performed a new sequencing totally dedicated to mtDNA, enriched by hybridization capture with a Neanderthal bait panel (Arbor Biosciences). According to this new analysis, the ES1, ES2, ES5 and Xb samples from SU X were positive for the presence of ancient human mtDNA. Please, see the new Figure 2B, Table S3, L107-116 and 354-379 for the new methods.

In conclusion, we believe we have the following evidence to support the *Homo* origin of the fecal remains of the EI Salt samples (especially those from SU X):

- The presence of several millimetric phosphatic coprolites and fecal lipid biomarkers - namely coprostanol and 5-beta-stigmastanol – with proportions suggesting a human origin;
- The presence of human mtDNA traces in PMDS-filtered metagenomic reads and, vice versa, the absence of any other animal mtDNA;
- Confirmation of the presence of human mtDNA through target capture and sequencing;
- The presence of ancient metagenomic traces of microorganisms of possible human origin.

For one, species identification (especially from ancient DNA) can be difficult if the proper reference genomes are not present -- reads can be assigned sub optimally or erroneously due to database bias (see Figure 4 from Warinner et al. 2017: A Robust Framework for Microbial Archaeology. *Annu. Rev. Genom. Hum. Genet.* <https://doi.org/10.1146/annurev-genom-091416-035526>). The veracity of the microbial species assignments by the authors must be tested using edit-distances (see Hubler et al. 2019: HOPS: automated detection and authentication of pathogen DNA in archaeological remains. *Genome Biology*. DOI: 10.1186/s13059-019-1903-0).

We totally agree with the Reviewer.

Following her/his suggestion, the veracity of the microbial species assignments has been strengthened by applying the approach recently used by Jensen et al. (2019 doi: 10.1038/s41467-019-13549-9), which is essentially based on the HOPS pipeline. In particular, all the reads retrieved from the samples IX, Xa, ES1-7, Xb and XI (i.e., those positive for the presence of fecal biomarkers and/or associated with rich archaeological assemblages) were first annotated and, subsequently, the ancient origin of each taxon was authenticated by computing 3 parameters: fraction of reads with PMDS >1, and $-\Delta\%$ and 5 deamination rate of PMDS >1 reads. Data from this new analysis are provided in the new Table 1 and as Supplementary Figures S3-S5 (MapDamage plots, coverage plots and edit distance distribution). Based on these new results, we were able to confirm the presence of several species belonging to the GM families of hominids, showing a read profile consistent with their ancient origin, as discussed in L177-187. Please, see L331-352 for new methods.

Overall, I think that the authors propose an interesting idea, but more work is needed for it to be robust and convincing.

We really hope that the additional work we have done will make our work more robust.

Minor comments:

Lines 267-268: "entire set of bacterial genomes of the NCBI database (downloaded on November 15th, 2017)"
The authors may wish to be a bit more specific here, as there are multiple sets of bacterial genomes available at the NCBI, e.g. RefSeq, GenBank, etc. Also, I presume that their selection includes Archaea?

We apologize for the inaccuracy.

"The entire set of bacterial genomes of the NCBI database" we were referring to included RefSeq entries. Archaea were included as well. These details have now been added to the text (L300).

Figure S1, which depicts DNA deamination plots is a bit confusing to me for a couple of reasons.

1. Typically, such plots depict the alignment of a sample's reads to a single reference genome. How is this done for the metagenomic samples in this paper, are they summed across all microbial species?
2. The methods state that the authors used a UDG-half treatment, but the plots in this figure do not look like they have had this treatment. UDG-half treatment should drastically reduce the deamination signal observable to the last few terminal bases. (see figure 2 from (Rohland et al. 2015: Partial uracil-DNA-glycosylase treatment for screening of ancient DNA. *Phil. Trans. R. Soc.* B37020130624 <http://doi.org/10.1098/rstb.2013.0624>)

We apologize for the lack of clarity.

Regarding the first point, the plots were generated for each sample by summing across all microbial species.

As for the second, the figure refers to reads with PMDS >5 and the plots reflect the typical trends of ancient samples. Anyway, as correctly noted by the Reviewer, since through the UDG-half treatment, deaminated bases are partially removed and retained mostly at the ends of the molecule, the new approach by Jensen et al. (i.e., edit distance distribution and MapDamage plots) was specifically applied by considering the first 10 bases of reads with PMDS >1.

Reviewers' comments:

Reviewer #1 (Remarks to the Author):

I believe the authors have made a considerable amount of work to improve the study. Specifically, the addition of the mtDNA sequencing strengthens the study. The authors also seemed to have covered any other concerns, add literature, and new interpretations as needed. I believe that the manuscript is now suitable for publication.

Tasha M. Santiago-Rodriguez

Reviewer #2 (Remarks to the Author):

The addition of the Neanderthal mtDNA analysis (alongside the other additional checks that were done) is a very nice addition and adds a lot of credibility to the study. However, in the legend of Figure 2B, it suddenly states the following: "Red boxes, mtDNA fragments of Homo sapiens as recovered from metagenomic sequencing data; red circles, confirmation of the presence of ancient human mtDNA through target capture and sequencing (please, see methods for further details)." Homo sapiens is not Homo neanderthalensis. Is there a mistake in the legend? Can you specifically identify particular SNPs (single nucleotide polymorphisms) in the ancient hominid mtDNA (after filtering) that clearly identify (some of) these mtDNA reads as coming from Homo neanderthalensis and not from Homo sapiens? Perhaps show/visualize some of these SNPs (showing both reference genomes + reads)?

I kind of like the figures made in response to reviewer 1 (Figures 1 and 2). Of course, you can't take the numbers (relative abundances) at face value as you say due to various factors; but simply just say so in the legend or in the text. The figures themselves, even with this caveat look informative (perhaps make them supplemental). I would strongly recommend including the comparison populations in Figure 1 as well. Also, perhaps similar bar chart figures of the other soil layers as well (to show that these are very different / mostly environmental). Perhaps, just as a visualization of the contamination removal process, perhaps also show the profile before screening for ancient DNA (before and after picture in essence).

"Lines 164-169: Can the authors include the relative abundances for each bacterial genus?

Lines 180-185: Can the authors include the relative abundances for each bacterial species?

As discussed above, if the Reviewer agrees, we would prefer not to include this information but to limit the presentation of the results to the distribution of the number of hits as shown in Figure 4."

I would second the request of reviewer 1. Of course the numbers should be taken with a pinch (or a bucket) of salt, but the composition nonetheless already looks credible.

In regards to Spirochaetes (Treponema), the authors do seem to have a point that the Treponema species found could perhaps indeed be oral because as you would expect them to be somewhat more similar to the "modern" hunter-gatherer gut Treponema species if they were themselves also gut species. I'm satisfied with their response in regards to this.

Minor comments:

Nevertheless, to the best of our knowledge, paleofeces older than 8,000 years ago have never being analyzed, leaving an important gap on the pre-historical human GM configuration.

- Perhaps change "have" to "has", or say paleofeces "samples" older than Also, remove "ago" (you already have "older" in the sentence)

The archeological setting of El Salt yielded evidence of recurrent occupation by Neanderthals, our

closest evolutionary relatives, dated between 60.7 ± 8.9 and 45.2 ± 3.4 kya (26, 27).

- Modern Europeans are few % Neanderthal (Asians are also include some Denisovan). They should perhaps not be considered just relatives but also ancestors to a degree.

which were all detected, at variable but substantial abundance,

- abundance"s"?

Reviewer #2 (Remarks to the Author):

The addition of the Neanderthal mtDNA analysis (alongside the other additional checks that were done) is a very nice addition and adds a lot of credibility to the study. However, in the legend of Figure 2B, it suddenly states the following: "Red boxes, mtDNA fragments of Homo sapiens as recovered from metagenomic sequencing data; red circles, confirmation of the presence of ancient human mtDNA through target capture and sequencing (please, see methods for further details)." Homo sapiens is not Homo neanderthalensis. Is there a mistake in the legend? Can you specifically identify particular SNPs (single nucleotide polymorphisms) in the ancient hominid mtDNA (after filtering) that clearly identify (some of) these mtDNA reads as coming from Homo neanderthalensis and not from Homo sapiens? Perhaps show/visualize some of these SNPs (showing both reference genomes + reads)?

The Reviewer is right, in fact there was a mistake in the legend of Figure 2B, which was somehow misleading. In the revised version of our manuscript, we changed the legend as follows (please see L733-738):

(B) Red boxes, human mtDNA fragments as recovered from metagenomic sequencing data; red circles, confirmation of the presence of ancient human mtDNA through target capture and sequencing (please, see methods for further details).

Unfortunately, for the samples ES1, ES2, ES5 and Xb, which tested positive for the presence of ancient human mtDNA (through target capture and sequencing), the coverage and breadth of coverage of mtDNA did not allow discrimination between *Homo sapiens* and *Homo neanderthalensis* and, for this reason, in the text we always refer to ancient human mtDNA.

I kind of like the figures made in response to reviewer 1 (Figures 1 and 2). Of course, you can't take the numbers (relative abundances) at face value as you say due to various factors; but simply just say so in the legend or in the text. The figures themselves, even with this caveat look informative (perhaps make them supplemental). I would strongly recommend including the comparison populations in Figure 1 as well. Also, perhaps similar bar chart figures of the other soil layers as well (to show that these are very different / mostly environmental). Perhaps, just as a visualization of the contamination removal process, perhaps also show the profile before screening for ancient DNA (before and after picture in essence).

As suggested by the Reviewer, Figures 1 and 2 from the previous rebuttal are now provided in the Supplementary Information as fig. S3 and S5, respectively. The data are discussed in the text (L155-156, 158-167), obviously stressing all the possible biases and precautions to be taken for their interpretation. Furthermore, as suggested, a new supplementary figure, fig. S4, showing the compositional differences for PMDS > 5 reads between samples positive for the presence of fecal biomarkers and/or associated with rich archaeological assemblages (samples IX, Xa, ES1-7, Xb and XI), and samples with no or very few archeological remains (from SU V) is now provided and discussed (L156-158).

"Lines 164-169: Can the authors include the relative abundances for each bacterial genus?

Lines 180-185: Can the authors include the relative abundances for each bacterial species?

As discussed above, if the Reviewer agrees, we would prefer not to include this information but to limit the presentation of the results to the distribution of the number of hits as shown in Figure 4."

I would second the request of reviewer 1. Of course the numbers should be taken with a pinch (or a bucket) of salt, but the composition nonetheless already looks credible.

As indicated above, the relative abundances of components of the 24 families common to the gut microbiome of hominids in samples positive for the presence of fecal biomarkers and/or associated with rich archaeological assemblages (samples IX, Xa, ES1-7, Xb and XI) vs samples with no or very few archeological remains (from SU V) are now provided as new fig. S4 (L156-158).

In regards to Spirochaetes (Treponema), the authors do seem to have a point that the Treponema species found could perhaps indeed be oral because as you would expect them to be somewhat more similar to the "modern" hunter-gatherer gut Treponema species if they were themselves also gut species. I'm satisfied with their response in regards to this.

We are happy to have fulfilled the Reviewer's requests.

Minor comments:

Nevertheless, to the best of our knowledge, paleofeces older than 8,000 years ago have never being analyzed, leaving an important gap on the pre-historical human GM configuration.

- Perhaps change "have" to "has", or say paleofeces "samples" older than Also, remove "ago" (you already have "older" in the sentence)

The archeological setting of El Salt yielded evidence of recurrent occupation by Neanderthals, our closest evolutionary

relatives, dated between 60.7 ± 8.9 and 45.2 ± 3.4 kya (26, 27).

- Modern Europeans are few % Neanderthal (Asians are also include some Denisovan). They should perhaps not be considered just relatives but also ancestors to a degree.

which were all detected, at variable but substantial abundance,

- abundance"s"?

All minor changes indicated have been made.

REVIEWERS' COMMENTS:

Reviewer #2 (Remarks to the Author):

I'm satisfied with the responses and changes and I only have a few minor comments left:

Main Manuscript:

Lines 179-183: Bit of a long sentence. Could chop it up?

It is also worth remembering that most of these bacteria are able to produce short-chain fatty acids (SCFAs) (mainly acetate and butyrate), which are today considered metabolic and immunological GM players with a leading role in human physiology, produced by the microbial fermentation of indigestible carbohydrates, through the establishment of complex syntrophic networks (48).

Lines 228-232: Bit of a long sentence. Could chop it up?

Expectedly, the samples from the upper part of SU V (which are poor in archaeological remains) showed scarce and inconsistent presence of aDNA related to hominid-associated GM bacterial families, with the highest hit counts for *Clostridium perfringens*, *Paeniclostridium sordellii* and *Turicibacter sanguinis*, the first two being environmental opportunistic microorganisms historically associated with human gangrene and the last with acute appendicitis (68-70).

Legend Figure S2: "PMDS >5"; remove the space between S and > (like it is on the Y-axis, or change the Y-axis to the legend, or change both to something identical). Make this consistent throughout the manuscript (like in legend of Fig. S4 for example, or the other way around you have "PMDS<1" in the legend of Fig. S8).

Figure S3: Y-axis: Fix the spelling of "realative". Does "Abundance" really need a capital letter? (its not German)

Figure S7: Curiosity question: Why is the coverage for *C. perfringens* so "spikey" (and *P. sordelli*)? If known, perhaps add a note to the legend?

REVIEWERS' COMMENTS:

Reviewer #2 (Remarks to the Author):

I'm satisfied with the responses and changes and I only have a few minor comments left:

Main Manuscript:

Lines 179-183: Bit of a long sentence. Could chop it up?

It is also worth remembering that most of these bacteria are able to produce short-chain fatty acids (SCFAs) (mainly acetate and butyrate), which are today considered metabolic and immunological GM players with a leading role in human physiology, produced by the microbial fermentation of indigestible carbohydrates, through the establishment of complex syntrophic networks (48).

The sentence was broken in 2, as indicated.

Lines 228-232: Bit of a long sentence. Could chop it up?

Expectedly, the samples from the upper part of SU V (which are poor in archaeological remains) showed scarce and inconsistent presence of aDNA related to hominid-associated GM bacterial families, with the highest hit counts for *Clostridium perfringens*, *Paenibacillus sordellii* and *Turicibacter sanguinis*, the first two being environmental opportunistic microorganisms historically associated with human gangrene and the last with acute appendicitis (68-70).

As suggested, this sentence was also split into 2.

Legend Figure S2: "PMDS >5"; remove the space between S and > (like it is on the Y-axis, or change the Y-axis to the legend, or change both to something identical). Make this consistent throughout the manuscript (like in legend of Fig. S4 for example, or the other way around you have "PMDS<1" in the legend of Fig. S8).

Thank you so much for the correction. The use of spaces has been revised throughout the manuscript.

Figure S3: Y-axis: Fix the spelling of "realative". Does "Abundance" really need a capital letter? (its not German)

Thanks for the correction that was made.

Figure S7: Curiosity question: Why is the coverage for *C. perfringens* so "spikey" (and *P. sordellii*)? If known, perhaps add a note to the legend?

We thank the Reviewer for this comment.

Indeed for some regions of the genome of those microorganisms we got a greater coverage.

It is hard to say why. We might speculate that these are more conserved regions than others, perhaps common to several bacteria (as a result of horizontal gene transfer). Obviously, this remains only a hypothesis, not verifiable in the course of our study (unfortunately it was not possible to reconstruct the entire genome). In light of this, we would prefer not to add any notes in the legend.